# Digitally Sustainable Information Systems in Axiomatic Design

**Fabrizio Pecoraro [1], Elaheh Pourabbas [2], Fernando Rolli [3] and Chiara Parretti [3,\*]**

[1] Institute for Research on Population and Social Policies, National Research Council,
32-00185 Rome, Italy; f.pecoraro@irpps.cnr.it
[2] Institute for System Analysis and Computer Science "Antonio Ruberti", National Research Council,
19-00185 Rome, Italy; elaheh.pourabbas@iasi.cnr.it
[3] Department of Engineering Science, Guglielmo Marconi University, 44-00193 Rome, Italy;
f.rolli@unimarconi.it
\* Correspondence: c.parretti@unimarconi.it; Tel.: +00-393-204-267-676

**Abstract:** Nowadays, information systems are evolving towards increasingly interconnected, smart, and self-adaptive models. This transformation has led to the representation of the systems themselves in terms of natural ecosystems. Similar to the natural environment, the virtual world can be threatened by specific forms of pollution, such as illegitimate access to the system, unwanted changes to data, and loss of information, which affect the only resource it possesses, i.e., data. In order to provide proactive protection of data integrity and confidentiality, in this paper we consider the well-known principles of privacy by design and privacy by default in the design phase of system development. To this end, we propose an approach based on axiomatic design, which allows us to implement these two principles through an appropriate reinterpretation of the information axiom, in terms of privacy impact assessment. We illustrate our approach by a case study, which implements the process of managing patients in home care. However, the proposed method can be applied to processing systems that provide services. The main result achieved is to select the most digitally sustainable design solution, i.e., the one that best prevents the threats mentioned above.

**Keywords:** digital sustainability; axiomatic design; privacy impact assessment; privacy-by-design; privacy-by-default

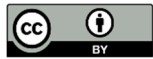

## 1. Introduction

In the1960s, the neoclassical paradigm of economic development came into sharp crisis. The scientific and political debate that followed has focused on the limits of development based on the indiscriminate exploitation of natural resources. The turning point of this change was the publication in 1971 of the essay "The Entropy Law and the Economic Process" by N. Georgescu-Roegen [1]. Georgescu-Roegen was the first economist to adapt the laws of thermodynamics to economic processes. In particular, the second principle of thermodynamics allowed him to theorize that economic processes also involve a degradation of the resources used to produce goods and services. Therefore, he deduced that future generations will have fewer resources and hence fewer opportunities and rights than present generations [1]. The debate in the intervening years has moved from academic classrooms to the political agendas of governments. In 1987, the United Nations published the so-called Brundtland Report (also known as Our Common Future). It represents the first global position on environmental issues. The Brundtland Report introduced the general term of sustainable development, understood as "meeting the needs of today without compromising the ability of future generations to meet their own needs" (The Brundtland Commission, 1987)" [2]. This concept, however, goes beyond the immediate economic and environmental considerations of sustainability itself [3].

Genuine sustainable development has three dimensions: economic, environmental, and social. The first dimension is related to the soundness of economic processes, while the second is related to the preservation of environmental resources. The third relates to the development of an inclusive society, which offers all its members opportunities for personal fulfillment, well-being, and health protection [4]. In this paper, we focus on this last dimension of sustainability, with particular reference to the design of information systems. They ensure the delivery of multiple utility services to citizens, which, similar to those related to telemedicine and home care support, are particularly relevant in this era of the COVID-19 (Coronavirus Disease 2019) pandemic. In the literature, there are not many studies that explore social sustainability issues in information system design [5]. This is due to the fact that information communication technology (ICT) is predominantly considered a tool of social sustainability since in every sector it has made possible the improvement of the quality of life of citizens [6,7]. However, as some authors have pointed out, information systems can also be considered ecosystems [5,8–10]. In these cases, the true wealth of an information system is represented by the data it maintains [9,11,12]. Similar to natural ecosystems, information systems can also be, voluntarily or involuntarily, subjected to threats that can degrade their resources [5]. From this point of view, an information system can be defined sustainable if it is designed in such a way as to preserve the integrity and confidentiality of the data. In the field of personal data protection, in the '90s, Anna Cavoukian [9] introduced the principles of privacy by design and privacy by default, which despite being conceived for systems with a low level of interoperability, are even more effective with current architectures, which provide for an elevated interchange of data between different platforms (Internet of Things, Mobile Systems, Cloud, Artificial Intelligence, Big Data). In this new context, the adoption of these principles is equivalent to pursuing the digital sustainability of an artificial ecosystem, but whose behavior is not dissimilar in operation from a biological system [10]. They aim to preserve the only resource available to these artificial environments, i.e., data. In fact, the principle of privacy by design consists of including, already in the design phase, all of the necessary measures to minimize the occurrence of this type of threat. The concept of privacy by default involves the definition of predefined ways of using the system in order to avoid possible data breaches.

In this paper, we propose an approach based on axiomatic design, which allows us to implement these two principles through an appropriate reinterpretation of the information axiom, in terms of a Privacy Impact Assessment (PIA). Axiomatic design is particularly appropriate for operating in complex environments since it makes use of a pragmatic approach in solving individual cases based on a view of complexity in terms of the functional behavior of the system [13]. It allows us to identify for each operational step a finite set of alternative design solutions, which are obtained from the decomposition of the functional requirements of the system [14]. At this point, we proceed to the selection of the robust solution, which results from an assessment of the impact of possible threats that the system must avoid to be considered sustainable. In this paper, we propose the adoption of a comparative methodology of the PIA type [15], which allows ordering the set of functionally admissible solutions with respect to three specific categories of risks, which are, specifically, attempts of illegitimate access to the system, unwanted changes to data, and loss of information. These threats are explicitly stated in current legislation, which in the context of the European Union refers to the General Data Protection Regulation No. 2016/679 (GDPR) [15]. This regulation takes into account the ongoing technological evolution, introducing a series of new rules on the processing of personal data. First, clearer rules are defined for information and consent, which become mandatory to allow the processing of citizens' data. In addition, the criterion of the minimum scope of personal data is introduced, according to which only the information necessary to allow the requested service to be carried out should be requested from citizens. The collection and management of data, which is not relevant to the requested processing, becomes an explicit violation of the GDPR. The foundations are also based on

new rights such as the right to erasure of personal data, the "right to be forgotten" on Internet search engines, and the right to block the processing of personal data [15].

As an example, we report a case study regarding the implementation of an integrated home care system for partially self-sufficient patients. The application of the proposed approach allows selection of the most appropriate design solution to prevent the three threats previously introduced. In this perspective, the concept of digital sustainability is equivalent to the design of a system that does not allow in any way the patient's information to be altered, disclosed or processed for purposes other than those intended by the specific treatment [15]. However, this approach finds application in the general field of information systems design, for which breach actions on the data being processed have a direct impact on the lives of exposed subjects or may constitute serious conditioning of citizens' opinions.

Finally, it is also necessary to consider that from an economic point of view, the data protection sector is growing rapidly. A recent study performed in [16] showed that the global size of this market was estimated at \$9,532.16 million in 2021. It is expected to grow to \$27,923.84 million by 2027. Consequently, our study aims to present a new methodological approach for solving real-world design problems in a rapidly growing sector.

The paper is structured as follows. The next section presents the underlying axiomatic design methodology. In Section 3, the privacy impact assessment in axiomatic design is discussed. In Section 4, a case study regarding integrated health and social records is presented, where the application of privacy impact assessment is shown. Section 5 concludes.

## 2. Axiomatic Design Approach: Background

In axiomatic design, the first step in developing a project is to correctly define the problem to be solved [17]. It allows us to identify the functional requirements, which represent the design formalization of customer needs (*CA*) (see Figure 1). This is a crucial activity, which often other engineering approaches tend to underestimate, jumping too hastily to the proposal of technical solutions [18], which sometimes can be inadequate to the specific context. On the contrary, axiomatic design allows correspondence between the customer needs and the proposed solutions. This correspondence can be subject to decomposition up to minimum levels of detail, providing the design artifact with a hierarchical representation in terms of design matrices (DM), or in equivalent modular diagrams (module–junction structure diagram, design flow diagram, unified modelling language diagrams) [19].

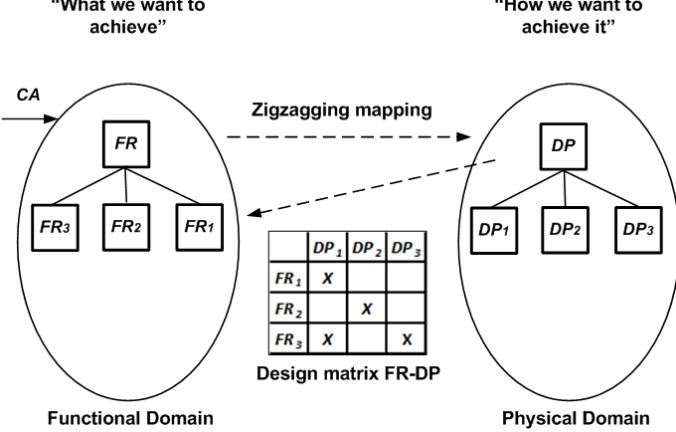

**Figure 1.** Axiomatic design high-level paradigm representation.

For each decomposition, we build a mapping between functional requirements (*FRs*) and ways of implementing them, called design parameters (*DPs*) [14]. In a simplified

form, we can define at least two domains (see Figure 1), the functional domain and physical domain. The former represents the "what to do", and the latter represents the "how to do" domain.

In this context, axiomatic design makes it possible to define a mapping between these two domains in terms of a design matrix (Figure 1). In this case, the "*X*" symbols represent the mapping between *FRs* and *DPs* and identify the constituent modules of the system [19]. For this reason, we can say that the design matrix is a particular domain mapping matrix (DMM), because it maps the relationships that exist between two domains [20]. Usually, DMMs are *nxm* rectangular matrices that relate the *n* elements of one domain to the *m* elements of another domain. Conversely, design matrices are always square matrices. This consideration introduces us to the most specific feature of axiomatic design, i.e., the independence axiom.

This axiom states that the admissible design configurations are only those that guarantee that the functional requirements remain independent [14]. This means that the design matrices representative of acceptable relationships are square matrices of diagonal (uncoupled) or triangular (decoupled) types. The other matrix forms are equivalent to inter-domain relationships that have mutually dependent functional requirements (coupled). For this reason, axiomatic design involves a process of interdomain functional decomposition (zigzagging) that ensures that the design matrix is always a square matrix (Figure 1) [18,20]. From a conceptual point of view, the independence axiom guarantees the independence of the functional requirements that, upon the introduction of a perturbation ε on a single module, the response of the system remains predictable, i.e., observable. In fact, in the case of an uncoupled design matrix, the perturbation *ε* remains limited to the single perturbed module. Thus, the system response corresponds to the sum of the constituent modules, including that of the perturbed module. The same observation continues to be valid sin the case of system having a decoupled design matrix. In this case, the relationships between the various modules of the system are well defined because each module is connected to the next in an "in series" mode, i.e., the input of a module corresponds to the output of the previous one [19]. Instead, in the case of a coupled design matrix, the predictability of the system response is reduced. The modules are interconnected with feedback relationships. It becomes very complicated to determine the overall response of the system. This situation becomes more and more complex as the number of mutual interconnections (interfaces) between modules of the project matrix increases [21].

Instead, the information axiom allows us to select from a finite set of independent solutions the least complex design configuration. This second axiom states that the solution defined as robust is the one with the least information content, which is formalized as follows [22]:

$$I_i = -log\left(\frac{1}{P_i}\right); \tag{1}$$

where $P_i$ is the probability that the selected design configuration satisfies the *i*-th functional requirement (*FRi*).

Generally, at least three levels of decomposition can be defined for information systems [19]. The highest level of abstraction corresponds to the conceptual design of the system. At this level, functional requirements are formalized as use cases, while *DPs* are the mutual interactions (collaborations) [23–25]. In this case, the design matrix representative of the inter-domain mapping can be expressed as a collaboration diagram. With successive decomposition, we can fragment the single cases of use in specific actions. In other words, we can describe the dynamic behaviour of the system as a sequence of actions. This level corresponds to the logical design of the system. The corresponding design matrix can be translated into a UML (Unified Modelling Language) sequence diagram [23,25]. The other level of functional decomposition introduces us to the physical design of the system. In this case, the design matrix represents the mapping between

elementary functions (methods) and sets of data. This representation is equivalent to a UML class diagram [19]. At this point, it is possible to proceed to implement the system.

### 3. Case Study: Integrated Health and Social Care Record

This section describes a case study taken as an example for the design and implementation of an integrated health record system (longitudinal electronic health record). It comprises the access of both health and social care services and thus requires access to the electronic health record (EHR) as well as to external health care systems and social care records. In particular, the case study is centred on the data collection perspective with the aim of designing an interoperable integrated system. It specifically describes a real situation where a patient with limited self-sufficiency requires access to multiple services to monitor his/her daily life and to define specific therapeutic and social assistance plans. However, the integration of data from heterogeneous systems raises the question of identifying a design solution that preserves data integrity and confidentiality, i.e., ensures the digital sustainability of what is de facto an autonomous ecosystem. To define the high-level requirements of the system, we adopt the case story methodology [26,27]: M. P., 75 years old, female, partially self-sufficient following a cerebral ischemia and obesity. She lives alone. She is followed and checked periodically by medical staff (e.g., general practitioner and a specialist control centre made up of a cardiologist, neurologist, dietician, etc.) as well as by social workers that provide services to support her daily life needs (e.g., house cleaning, shopping, bill payment, social life). The patient's clinical and social information is stored in several distinct databases: general practitioner medical record, specialist medical records, electronic health record within the hospital, and social record managed by social services at a municipality level. The main aim of the integrated system is to provide a suitable solution to collect data from these systems and to show them to patient and providers according to their specific profile. The scenario described above is summarized in Figure 2, using the UML use case diagram that highlights actors involved in the care of a patient with complex care needs, as well as the information systems accessed by these actors during the provision of health and social care services. In particular, each actor provides a specific service to the patient (i.e., primary care, specialist and social care). Data and documents produced during each encounter between the patient and the provider are collected both in the provider specific system and in the integrated electronic health record (IEHR). This record provides to the professionals an interface to the patient's data depending on their specific role and responsibilities.

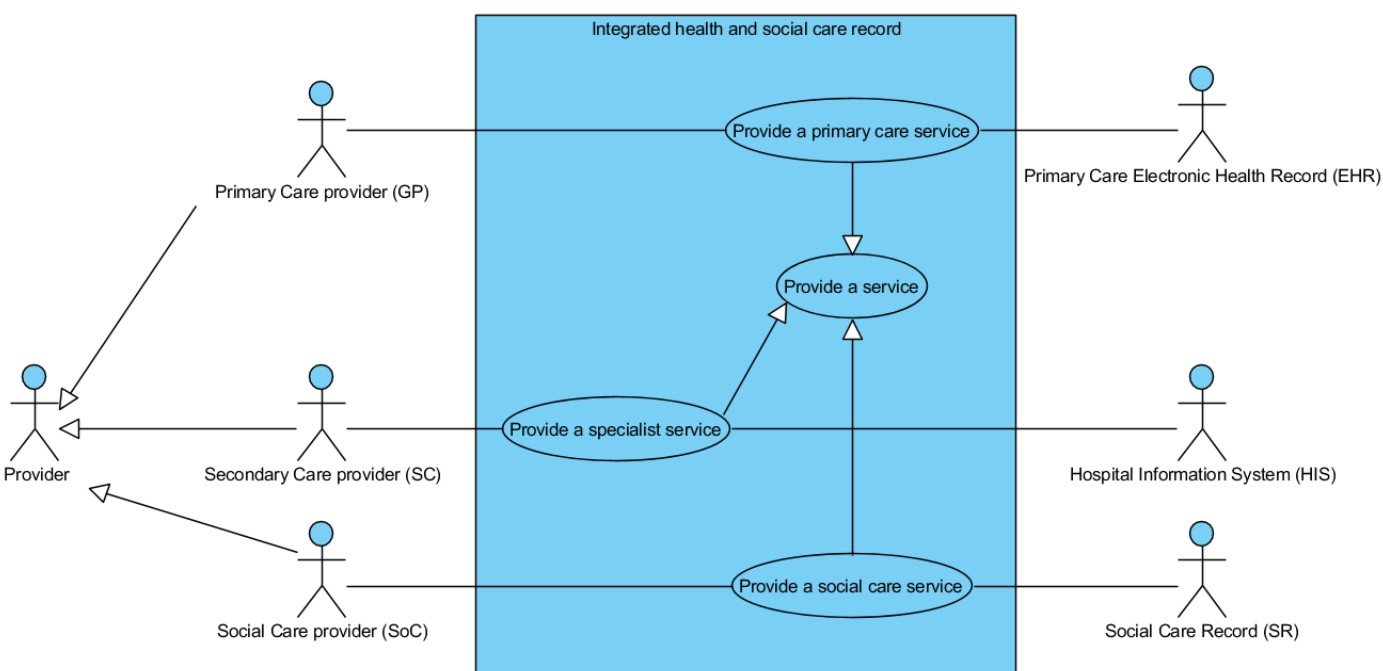

**Figure 2.** High-level UML use case diagram.

As highlighted in Figure 2, the integrated system collects data enclosed in three electronic records:

- the primary care record managed by the general practitioner who is in charge with the patient (electronic health record, EHR),
- the specialist record managed by the secondary care center placed in the hospital (hospital information system, HIS),
- the social record managed by the social worker and team placed in the referring territorial municipality (social record, SR).

Each information system is accessed during the provision of specific services represented here using relevant use cases, which are as follows: Provide a primary care service, Provide a specialist care service, Provide a social service. At this level of abstraction, they represent the functional requirements (*FRs*) of the system to be designed and implemented. Axiomatic design allows us to associate with each *FR* a specific implementation method (*DP*). Considering the case study described above and represented in Figure 2, the axiomatic design approach defines at least three alternative design solutions satisfying the independence axiom. They are shown in the three design matrices reported in Figure 3, where the rows represent the three use cases (*GP^j* = Provide a primary care service, *HP^j* = Provide a specialist care service, *Soc^j* = Provide a social service service), and the columns specify the interaction between them ($DP_i^j$), where *j* denotes the relevant solution. In this perspective, if the use case placed in column *i* interacts with the use case placed in column *j* (through a read and/or write action) the correspondence cell is flagged by the symbol "*X*" [25]. It is important to note that each actor may access and modify only the system he/she is in charge with, while other systems may be read depending on the type of data [28]. If the actor is allowed to read relevant-privacy data from other systems, then the corresponding cell is flagged, otherwise it remains empty.

| First matrix representation ($j=1$) | | |
|---|---|---|
| | $DP_1^1$ | $DP_2^1$ | $DP_3^1$ |
| $GP^1$ | X | | |
| $HP^1$ | X | X | |
| $Soc^1$ | X | X | X |
| Second matrix representation ($j=2$) | | |
| | $DP_1^2$ | $DP_2^2$ | $DP_3^2$ |
| $GP^2$ | X | | |
| $HP^2$ | X | X | |
| $Soc^2$ | X | | X |
| Third matrix representation ($j=3$) | | |
| | $DP_1^3$ | $DP_2^3$ | $DP_2^3$ |
| $GP^3$ | X | | |
| $HP^3$ | X | X | |
| $Soc^3$ | | | X |

**Figure 3.** Alternative design matrices corresponding to the high-level UML use case diagram.

In healthcare, the data being processed are particularly sensitive. Referring to the general data protection regulation (GDPR), we can identify the following threats that are considered extremely critical [29–34]:

- Illegitimate access to data ($C_1$),
- Unwanted data changes ($C_2$),
- Loss of data ($C_3$).

Therefore, the design of an integrated electronic health record (EHR) must consider these risks. In axiomatic design, we can reformulate these three threats in terms of internal input constraints because they are imposed by European data protection legislation [17]. This assumption allows us to guide the selection process towards the most digitally sustainable solution in order to minimize the data breach risks. Furthermore, the preventive application of these design constraints for every decomposition level involves the definition of a system, whose main modality of operation coincides with the base settings prescribed by privacy by default.

## 4. Privacy Impact Assessment in Axiomatic Design

Privacy impact assessment methodology has an interesting analogy with axiomatic design. Both approaches are based on a top-down decomposition of the processes/systems to be implemented, which then follow the re-engineered re-composition of the same according to a V model scheme [14,25]. This conceptual analogy lays the foundations for both methodologies to be used together. As we mentioned in Section 2, axiomatic design allows a modular representation of a system in terms of a design matrix or equivalent diagrammatic form. Admissible design solutions that satisfy the axiom of independence are represented by uncoupled design matrices, i.e., with independent modules. Moreover, they can be expressed by decoupled matrices having interconnections (interfaces), which define an unambiguous behaviour of the system in terms of sequence of actions. Only in these cases is it possible to design systems according to the principle of privacy by design. In the case of coupled systems, on the other hand, we are not able to predict the overall behaviour of the system, consequently making it impossible to provide an assessment of its level of digital sustainability. Therefore, a necessary condition for the use of axiomatic design as a tool for implementing the principles of privacy by design and by default is the perfect predictability or observability of the system to be designed. This specific condition logically constitutes validation of the proposed method. To introduce the PIA evaluation procedure, we start with considering the representation of a system in terms of a design matrix. In the previous section, we showed that the application of the independence axiom

can lead to the identification of a finite set of design matrices [21]. Each of these matrices corresponds to an admissible design solution. As shown in Figure 4, we need to select the most digitally sustainable solution, i.e., the one for which the risk of data breaches is minimal. To this end, we can resort to a particular reformulation of the information axiom. This axiom has the particularity that it can be easily contextualized to the operational environment to which it is to be applied. Thus, in our case, the concept of information content can be reinterpreted in terms of the potential impact of system data pollution with respect to the specific design solution available. Thus, Equation (1) can be rewritten in the terms of the following:

$$I_i = PIA_i, \tag{2}$$

where $PIA_i$ is the data violation risk resulting from choosing a $DP_i$ solution that enables the implementation of the requirement *FRi*. This interpretation of the information axiom allows us to implement the principle of privacy by design by considering threats to data integrity and confidentiality as constraints on the operation of the system. Therefore, the set of admissible solutions that respect the information axiom is ordered with respect to an assessment of the risk of breaches on the data maintained by the system. This approach is equivalent to selecting the most digitally sustainable solution. Figure 4 summarizes the application of the proposed method for each level of functional decomposition.

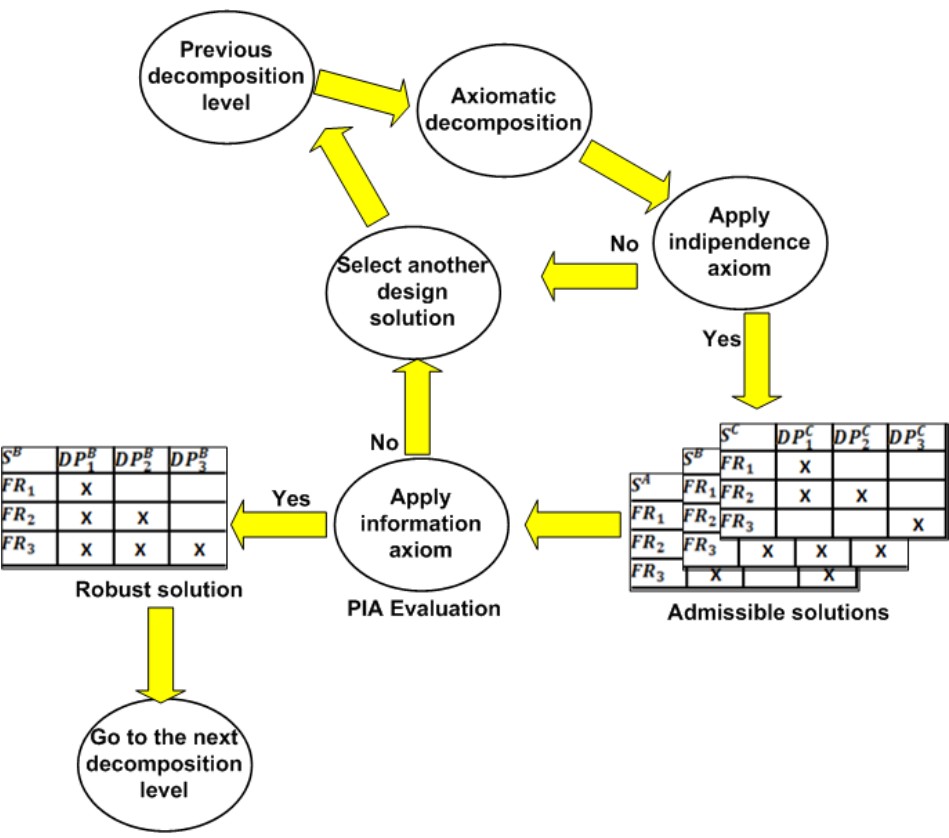

**Figure 4.** Process of selecting a digitally sustainable solution.

Each axiomatic decomposition cycle is defined as follows:

- At the "Previous decomposition level", the functional robust solution of the previous level is considered.
- At the "Axiomatic decomposition" step, we functionally decompose the robust solution of previous level, based on an inter-domain mapping process called zigzagging (Figure 1).

- At the "Apply independence axiom" step, only admissible solutions (S) are selected. They correspond to decoupled or uncoupled design matrices.
- At the "Apply the information axiom" step, the robust solution is selected. It corresponds to the solution with the best PIA evaluation.
- At the "Select another design solution" step, the robust solution of the previous level is replaced by another solution of the same level. This condition occurs if none of the decomposed solutions satisfies the independence axiom or if the impact assessment of the identified admissible solutions is not deemed acceptable.
- At "Go to the next decomposition level" step, the selected robust solution is subjected to further functional decomposition. This process is iterative and is continued at the level of detail deemed necessary.

The methodological approach represented in Figure 4 can be realized by performing a PIA evaluation for each admissible design matrix. For this reason, we define a comparison table called a privacy constraint matrix [35]. It is made by placing the $DP_i$ of the design matrices ($S_j$) to be evaluated along the rows, while the constraints ($C_k$) are placed along the columns (see Figure 5). Each $a_{i,k}^j$ element of the abovementioned matrix ($S_j$) represents the measure of the impact of the $C_k$ constraint on the $DP_i^j$ [25,35]. This value is determined according to Equation (3) as follows:

$$a_{i,k}^j = S_{i,k}^j * O_{i,k}^j, \tag{3}$$

where $S_{i,k}^j$ represents the seriousness of the occurrence of the breach of the $C_k$ constraint with respect to the $DP_i^j$ [29,30]; $O_{i,k}^j$ is the probability that the $C_k$ constraint is not respected by the $DP_i^j$ in Figure 3 [29,30]. $S_{i,k}^j$ severity is estimated in terms of entities of potential impacts on the patient resulting from non-compliance with the $C_k$ design constraint. The values $S_{i,k}^j$ are integers between 1 and 4, as shown in Table 1 [29,30]. The determination of these values can be done empirically on the basis of specific check lists [30].

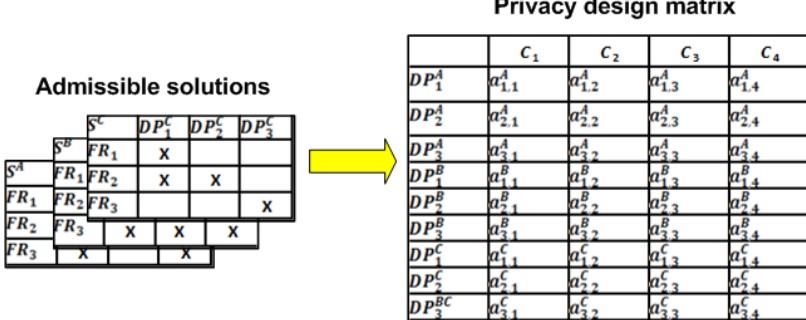

**Figure 5.** Privacy constraint matrix.

**Table 1.** Matrix for assessing the severity $S_{i,k}^j$ [29,30].

| Score | Description | Assessment |
|---|---|---|
| 1 | Negligible | The event did not cause any damage or caused minor inconveniences, such as the recompilation of some forms by the user. |
| 2 | Limited | The event caused temporary harm to the patient and necessitated additional interventions (additional costs, stress, temporary suspension of social services...). |
| 3 | Significant | The event caused temporary damage to the patient, which he/she should be able to overcome even if with serious difficulties (extension of hospitalization, withdrawal of social services...). |
| 4 | Maximum | The event caused permanent harm to the patient (permanent disability, long-term psychological or physical ailments, death). |

The $O_{i,k}^{j}$ probability represents the feasibility that the $C_k$ constraint could be breached. It is estimated primarily in terms of vulnerability level. The values of $O_{i,k}^{j}$ are integers between 1 and 4, as shown in Table 2 [29,30]. In this case, the determination of these values can be done empirically on the basis of appropriate check lists [30].

**Table 2.** Matrix for assessing the likelihood $O_{i,k}^{j}$ [29,30].

| Score | Description | Assessment |
|---|---|---|
| 1 | Negligible | There are no known events |
| 2 | Limited | Documented but not frequent |
| 3 | Significant | Documented and frequent |
| 4 | Maximum | Documented and very frequent |

Based on these assumptions, we can proceed to enhance the elements $a_{i,k}^{j}$ of the privacy constraint matrix (see Figure 5). The design solution corresponding to the design matrix that has the lowest overall value is selected. This assessment could also involve the redefinition of the functional requirements of the system in case no previously defined design solution has an acceptable value with respect to design constraints [25,35]. However, this does not pose any problem because axiomatic design is an iterative methodology, conceived to allow the reformulation of functional requirements or design constraints.

### 4.1. Application of Privacy Impact Assessment to the Case Study

The definition of the privacy constraint matrix can be carried out by adopting the PIA as the evaluation method. In this case, the $DP_i^{j}$ interactions introduced in Section 3 might be evaluated in a two-dimensional system that allows us to assess the severity of the possible violation as well as the probability when it occurs. The assignment of values is carried out by adopting an empirical approach as suggested by the CNIL through judgment of experts [30]. In this case, the design matrix with the lowest score represents the most digitally sustainable solution. As a first step, we have to estimate the severity of a possible violation of design constraints ($C_k$) due to interactions ($DP_i^{j}$) considering the matrices shown in Figure 3. For example, $DP_1^{1}$ refers to the design matrix *j* = 1 that corresponds to the operations that the family doctor can perform. Therefore, $DP_1^{1}$ has the following meaning for the following use cases:

- GP₁: $DP_1^{1}$ represents the set activities performed by the family doctor to store the patient's health data in the his/her record (EHR),
- HP₁: $DP_1^{1}$ represents the option that the family doctor can access (in a read mode) all of the patient's medical information produced by the specialized hospital centre and stored in the relevant system (HIS),
- Soc₁: $DP_1^{1}$ represents the option that the family doctor can access (in a read mode) all of the patient's social care information produced by the social worker and stored in the relevant system (SR).

At this stage of the methodology, we estimate the extent of a possible violation by the interaction $DP_1^{1}$ of design constraints ($C_k$) introduced in Section 3:

- Illegitimate access to data: $C_1$ consists of the prohibition for the data controller to access information that is unrelated to the correct performance of its activities. This situation would be a violation of the provisions of art. 18 of the GDPR [15]. In our case, the holder of the $DP_1^{1}$ treatment is the family doctor. He is entitled to view all of the patient's health data, including those produced at the Specialized Hospital Centre, but the family doctor has no title to fully access to his patients' social record (SR). For this reason, the design matrix *j* = 1 of Figure 3 would lead to the establishment of an integrated file that could allow the family doctor to access data for which the treatment is not legitimate (i.e., social care data). Empirically, a

significant level of risk (i.e., score 3) has been attributed to a possible violation of this type.

- Unwanted data changes: the $C_2$ constraint concerns the possibility that the family doctor carelessly modifies the system data. This risk is very low in an integrated EHR considering its efficient data backup and recovery mechanisms. On the other hand, the family doctor cannot carry out any operation that involves writing data in the specialist centre's folder (HIS) as well as in the social folder (SR) [28]. The data visualization alone excludes any possible hazards of this type. A score of 1 is therefore attributed.

- Loss of data: the $C_3$ constraint represents the possibility that operations performed by the general practitioner lead to the loss of data in the system. This risk is very low in an integrated EHR. In fact, in this case, the family doctor cannot carry out any operation that involves writing data in the specialist centre's record (HIS) and in the social record (SR). Thus, data visualization alone excludes any problems of this type. A score of 1 is therefore attributed.

Following this methodology, it is possible to define the severity matrix as shown in Table 3.

**Table 3.** Matrix for assessing the severity.

|  | $C_1$ | $C_2$ | $C_3$ |
|---|---|---|---|
| $DP_1^1$ | 3 | 1 | 1 |
| $DP_2^1$ | 3 | 2 | 2 |
| $DP_3^1$ | 2 | 2 | 2 |
| $DP_1^2$ | 3 | 1 | 1 |
| $DP_2^2$ | 3 | 2 | 2 |
| $DP_3^2$ | 2 | 2 | 2 |
| $DP_1^3$ | 1 | 1 | 1 |
| $DP_2^3$ | 1 | 1 | 1 |
| $DP_3^3$ | 2 | 2 | 2 |

Subsequently, the risk probability assessment matrix is constructed according to the same procedure already adopted to define the matrix in Table 3. Let us consider the $DP_1^1$ interaction as a reference to be adopted to evaluate the likelihood of the data violations described in PIA. In this case, the probability of a possible violation of data constraints ($C_k$) by the interaction $DP_1^1$ is as follows:

- Illegitimate access to data: the integrated EHR allows the family doctor to access data relating to the patient's medical record. Therefore, the likelihood of accessing this data is set to 3,

- Unwanted data changes: the overall likelihood of careless modification of system data is considered to be very low. Therefore, it arises as a negligible level of occurrence (i.e., 1),

- Loss of data: the likelihood of data processing associated with $DP_1^1$ that results in data loss is considered to be very low and is rated 1.

The result of this step of the methodology is the starting point to define the matrix for assessing the likelihood reported in Table 4.

**Table 4.** Matrix for assessing the likelihood.

|  | $C_1$ | $C_2$ | $C_3$ |
|---|---|---|---|
| $DP_1^1$ | 1 | 1 | 1 |
| $DP_2^1$ | 3 | 1 | 1 |
| $DP_3^1$ | 2 | 1 | 1 |
| $DP_1^2$ | 1 | 1 | 1 |
| $DP_2^2$ | 1 | 1 | 1 |
| $DP_3^2$ | 2 | 1 | 1 |
| $DP_1^3$ | 1 | 1 | 1 |
| $DP_2^3$ | 3 | 1 | 1 |
| $DP_3^3$ | 2 | 1 | 1 |

On the basis of Table 3 and Table 4, it is possible to define the first-level privacy constraint matrix shown in Table 5. This matrix represents the evaluation of the impact of privacy constraints ($C_k$) with respect to the interactions between use cases ($DP_i^j$). Each element of the matrix is calculated on the basis of Equation (3) (see Section 3). Furthermore, Table 5 allows us to select the most appropriate design configuration of privacy constraints. In this case, the design matrix $j = 3$ is the most digitally sustainable solution. Given the iterative nature of the process, each functional decomposition corresponds to an evaluation of the impact of privacy constraints. This evaluation can lead to a reconsideration of the upper-level settings (see Figure 4).

**Table 5.** First-level privacy constraint matrix.

|  | $C_1$ | $C_2$ | $C_3$ | Score |
|---|---|---|---|---|
| $DP_1^1$ | 3 | 1 | 1 |  |
| $DP_2^1$ | 9 | 2 | 2 | 26 |
| $DP_3^1$ | 4 | 2 | 2 |  |
| $DP_1^2$ | 3 | 1 | 1 |  |
| $DP_2^2$ | 3 | 2 | 2 | 20 |
| $DP_3^2$ | 4 | 2 | 2 |  |
| $DP_1^3$ | 1 | 1 | 1 |  |
| $DP_2^3$ | 3 | 1 | 1 | 16 |
| $DP_3^3$ | 4 | 2 | 2 |  |

*4.2. Consecutive Decompositions*

Starting with the design solution associated with the design matrix $j = 3$, we can carry out additional functional decompositions. For each of these decompositions, we apply the PIA approach introduced in the previous section. Axiomatic design also allows us to break down the design artefact into interconnected modules. It allows us to decompose a complex project into autonomous modules to be entrusted to design teams with specific specializations, which can operate remotely and with timescales established by a rigid schedule. Axiomatic design allows it to keep track of successive decompositions [19]. It permits coordination of design even in contexts of high operational complexity. In our case, for simplicity of exposition, we limited the decomposition to the mapping between the general practitioner and medical specialist (e.g., neurologist). In practice, we have defined in terms of a design matrix and UML diagrams the use cases relative to the visit of the patient by the general practitioner (GP3) and the medical specialist (HP3). The functional decomposition of the use cases was pushed to the point of defining the logical design of the GP3–HP3 subsystem. Table 6 provides a summary for three consecutive functional decompositions.

**Table 6.** Summary of the consecutive decompositions.

| First Level | Level of Decomposition | |
| --- | --- | --- |
| | **Second Level** | **Third Level** |
| Primary care visit (GP3) | General practitioner visit (A) | General practitioner visit (1) |
| | Referral (B) | Send Referral to EHR (2)<br>Send Referral to LEHR (2.1)<br>Send Referral stored to EHR (2.2)<br>Send Referral stored to General practitioner (2.3) |
| Specialist care visit (HP3) | Specialist Patient Info (C) | Get Patient Info from HIS (3)<br>Send Patient Info HIS to Specialist (3.1)<br>Get Patient Info from LEHR (4)<br>Send Patient Info LEHR to Specialist (4.1) |
| | Specialist visit (D) | Specialist visit (5) |
| | Specialist Referral (E) | Send Specialist Referral to HIS (6)<br>Send Specialist Referral to LEHR (6.1)<br>Send Specialist Referral stored to HIS (6.2)<br>Send Specialist Referral stored to Neurologist (6.3) |

Figure 6 represents the logical design of the GP3–HP3 subsystem in matrix form. The GP3 and HP3 use cases have been decomposed into gradually more elementary actions. The elements of the rows are the use cases of the system and the actions obtained for functional decomposition as reported in Table 6. These actions are activated for the use case GP3 by the general practitioner, while they are activated by the medical specialist for the use case HP3. The cells of the matrix are marked with symbol $X$ if the use case indicated on the column axis is able to activate a process capable of modifying the state of the target use case [19]. As can be seen, the design matrix in Figure 6 is constructed on an empirical basis, interpreting the description of end-user desires [23,26]. Our design process is iterative in nature [14]. Therefore, as a starting point, this solution can be considered valid because it respects the independence axiom since the related design matrix is triangular. With subsequent iterations, following the scheme summarized in Figure 4, we can proceed to improve the conceptual design of the system as well. However, the phenomenon related to sequential coupling will always remain. This is due to the fact that the activation of the use case $FR_{i+1}$ always depends on the previous $FR_i$. This is a structural type situation [36]. On the other hand, this type of sequential coupling can be neglected because the relationship between FR-DP is a decoupled type.

| | | | DP13 | | | | | DP23 | | | | | | | | |
| --- | --- | --- | --- | --- | --- | --- | --- | --- | --- | --- | --- | --- | --- | --- | --- | --- |
| | | | DPA | DPB | | | | DPC | | | | DPD | DPE | | | |
| | | | DP1 | DP2 | DP2.1 | DP2.2 | DP2.3 | DP3 | DP3.1 | DP4 | DP4.1 | DP5 | DP6 | DP6.1 | DP6.2 | DP6.3 |
| GP3 | A | 1 | X | | | | | | | | | | | | | |
| | B | 2 | X | X | | | | | | | | | | | | |
| | | 2.1 | | X | X | | | | | | | | | | | |
| | | 2.2 | | | X | X | | | | | | | | | | |
| | | 2.3 | | | | X | X | | | | | | | | | |
| HP3 | C | 3 | | | | | | X | | | | | | | | |
| | | 3.1 | | | | | | X | X | | | | | | | |
| | | 4 | | | | | | | X | X | | | | | | |
| | | 4.1 | | | | | | | | X | X | | | | | |
| | D | 5 | | | | | | | | | X | X | | | | |
| | E | 6 | | | | | | | | | | X | X | | | |
| | | 6.1 | | | | | | | | | | | X | X | | |
| | | 6.2 | | | | | | | | | | | | X | X | |
| | | 6.3 | | | | | | | | | | | | | X | X |

**Figure 6.** Matrix representation of the logic design of the GP3–HP3 subsystem.

Figure 6 can be easily represented using the UML sequence diagram as shown in Figure 7.

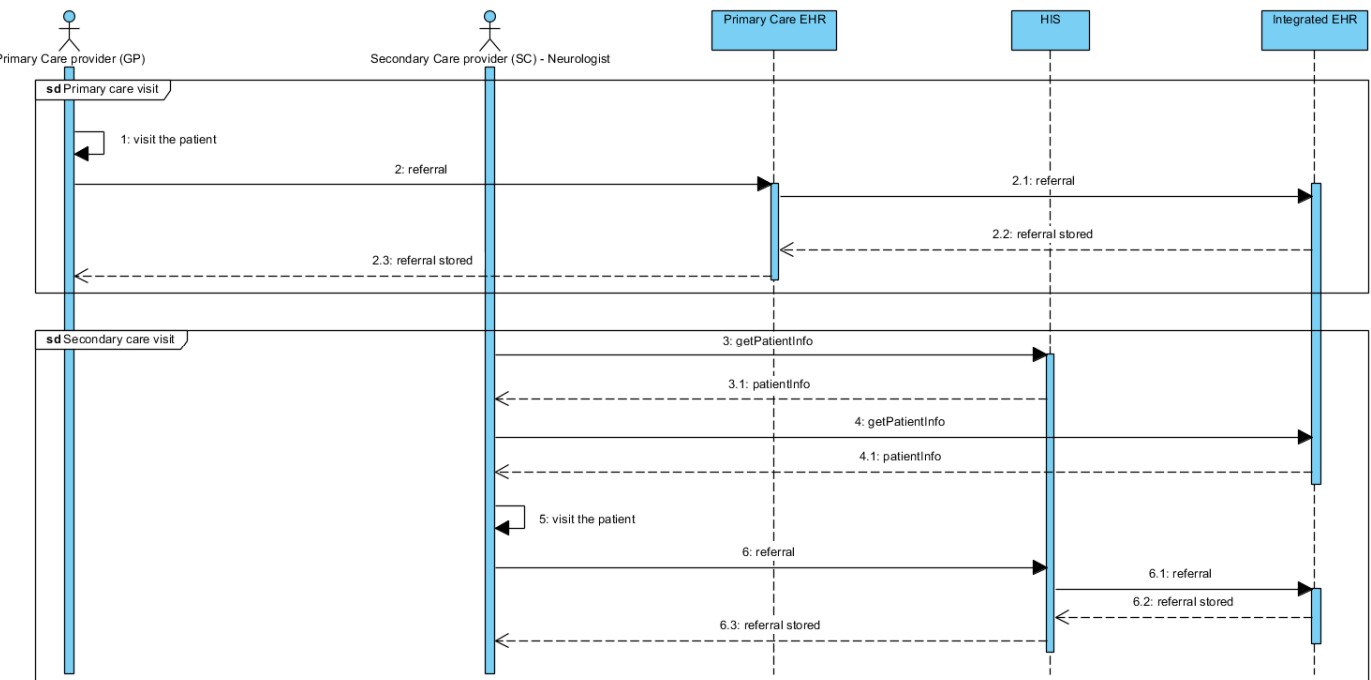

**Figure 7.** Sequence diagram of the GP3–HP3 subsystem.

As we have seen, axiomatic design allows us to define the logical design of the system, minimizing the risks of data pollution. This is an important result because it guarantees the digital sustainability of the whole system based on the preventive application of the privacy by design principle. In this case, data integrity and confidentiality protection measures are proactively included in the system design phase in terms of specific design constraints. This makes it possible to avoid costly post-implementation maintenance interventions or to expose patients to real risks of violating their privacy. At the same time, the last two representative forms (Figures 6 and 7) introduce us to the principle of privacy by design, which is a logical consequence of applying our method. Indeed, these two representative forms describe the main behaviour of the system. However, the adoption of a PIA-type methodology as a tool for implementing the information axiom has led to the selection of a design solution, whose main mode of operation is the one that least exposes the system to the risks of misuse by users. Therefore, this configuration of the system acts in a preventive way, limiting the occurrence of possible violations. Other ways of using the system are possible, but only as exceptions, to be verified and secured with tools provided by the GDPR (request for informed consent of the user, anonymization, encryption, and pseudo-anonymization of processed data) [31–34].

## 5. Conclusions

The major challenge of modernity is to make the use of ICT technologies socially sustainable. Technological evolution not only offers great opportunities but also raises important ethical issues concerning the treatment of the data that are processed. We have already highlighted how current systems are evolving towards models that are increasingly interoperable and self-adaptive, as much as that their behavior is not dissimilar from that of biological ecosystems, and just like natural environments, the virtual world can also be threatened by forms of pollution, which in this case concern the degradation of the only resource they possess, i.e., data. Data pollution is a looming threat because it can not only lead to the malfunctioning of a system, but for the spread of fake news on social networks, it can undermine the democratic foundations of civil communities, conditioning the choices of citizens.

We have proposed a case study concerning the processing of health data. However, the proposed design approach can be extended to other sectors that are particularly sensitive to data breach risks, such as platforms for electronic voting, e-commerce systems, social networks, customer loyalty systems of large distributions, etc. In this context, axiomatic design offers a powerful tool for implementing the principles of privacy by design and by default. This is possible in the first place because axiomatic design provides a modular representation of the design artifact. In this way, it is possible to provide an assessment of the impact of the individual modules that make up the system and their connecting interfaces inside and outside the system. In Section 2, we also pointed out that the independence axiom allows us to identify a finite set S of independent admissible design solutions. Only these types of solutions are admissible for digital sustainability evaluation as they exhibit predictable behavior. At this point, the information axiom, in the proposed reformulation in terms of PIA, allows us to select a robust solution. It is defined as the digitally sustainable solution, i.e., the one that best meets user needs and ensures compliance with data protection regulations. This result constitutes validation of the proposed methodology, demonstrating how axiomatic design and the principles of privacy by design and by default can be combined together in a particularly robust methodological approach. In this regard, it should also be noted that, in the field of information systems, the use of the information axiom is often overlooked. This problem stems from the fact that the standard definition of information content (I) is not readily applicable in software development projects. Alternative solutions have been proposed by resorting to redefining information content in terms of function points, user case points, or other measures of software complexity [24,37]. However, the proposed approaches have not had appreciable developments. Therefore, designers in this area often resort only to the application of the independence axiom. Similarly, the principles of privacy by design and by default have not found strong design tools that enable their implementation throughout all phases of the software lifecycle. Data protection authority guidelines in EU countries introduce them as general principles, without suggesting particular implementation methodologies. On the contrary, their integration with axiomatic design allows their effectiveness to be extended along all stages of software development. Moreover, the proposed approach makes it possible to provide a layered view of the same design artifact. It means that each level of design representation can be targeted to specific stakeholder categories. Design representation forms with higher abstraction can be addressed to decision makers, while those of higher detail can be considered for developers. This hierarchical articulation of the system allows for the logical integration of policy decisions, design choices, and implementation strategies, with an overall focus on transparency and sharing. In conclusion, we proposed a design approach, which combines the potential of axiomatic design to operate in complex environments, with the principles of privacy by design and by default, which were introduced in modern democracies to ensure the integrity and confidentiality of personal data. We believe that this approach proactively prevents the occurrence of data pollution situations, allowing the selection of the most digitally sustainable design solution, i.e., the solution with minimal risk of data pollution.

**Author Contributions:** All authors have contributed equally to the conceptualization, methodology, validation, formal analysis, investigation, writing—original draft preparation, writing—review and editing, visualization, supervision and project administration. All authors have read and agreed to the published version of the manuscript.

**Funding:** This research received no external funding.

**Institutional Review Board Statement:** Not applicable.

**Informed Consent Statement:** Not applicable.

**Data Availability Statement:** No new data were created or analyzed in this study. Data sharing is not applicable to this article.

**Conflicts of Interest:** The authors declare no conflict of interest.

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
