# Peer review of "Digitally Sustainable Information Systems in Axiomatic Design"

_sustainability, doi:10.3390/su14052598_

Round 1
Reviewer 1 Report
This paper is a good contribute to promote the coherency of social sustainability by the principles of axiomatic design.
Some issues that must be attended:
In Fig 1: where is “Design matrix FP-DP” must be “Design matrix FR-DP”
The legend of Fig 2 must be write with lower letters according with the others legends.
In the test appear "Fig. 1" and "Figure 1", it must be uniformized.
The sequence of call for figures don´t correspond to the same of the figures into the text; it must be revised.
In the Conclusions, the effect of application of Axiomatic Design must be improved.
Author Response
Dear Reviewer
thank you for your comments. We are uploading the revised manuscript and our point-by-point response to the comments.
Best regards
Chiara Parretti et al.
Responses to Comments
1) In Fig 1: where is “Design matrix FP-DP” must be “Design matrix FR-DP”.
Done
2) The legend of Fig 2 must be write with lower letters according with the others legends.
Corrected
3) In the test appear "Fig. 1" and "Figure 1", it must be uniformized.
Done
4) The sequence of call for figures don´t correspond to the same of the figures into the text; it must be revised.
The sequence of call for figures has been revised and corrected.
5) In the Conclusions, the effect of application of Axiomatic Design must be improved.
The conclusion has been extended and the application of Axiomatic design and our proposed method to other domains is discussed.
Reviewer 2 Report
The use of Axiomatic Design (AD) to evaluate privacy through appropriate interpretation of the information axiom is undoubtedly an interesting application of the theory to a novel area of application (information systems design). The basis for AD is well but superficially explained. The way is explained how the independence between Function Requirements (FR) is achieved via design matrices, which are then the basis for the system design. The design matrices should be design structure matrices (DSM), the solution finding in the physical domain (DM) is done by coupling the FR with the PD, but these are then domain mapping matrices (DMM). The illustrative pictures reduce this aspect, which makes it difficult to understand. In addition, the setting up of the SM appears to be the more critical point precisely because of the demand for independence between the functions, but this is neither explained nor justified. Instead, the reader is confronted with highly summarised information in the matrices. This seems counterproductive for the applicability or comprehensibility of the approach.
The interpretation of the information axiom as a description of private impact assessment, on the other hand, is in principle a very interesting and promising idea, but is difficult to interpret because of the lack of context. AD requires that functional structures are created (down to the level of detail) whose functions are largely independent of each other in order to reduce the information content. In terms of development methodology, this corresponds to the creation of modules, whereby modules are to be identified in such a way that they have as few or no interfaces as possible. Exactly this idea seems interesting for aspects of privacy, but is lost. The evaluation, i.e. the quantification ultimately as a decision support, only seems helpful if the underlying mechanisms are clear. Here I would recommend the authors to derive the method in more detail and to go into all steps.
The example is well prepared and contributes to understanding. A validation of the method is missing. Showing this on the same example, which is also used for the derivation, is not quite clean from a methodological point of view.
In principle, however, the paper is methodologically well structured and logically clear. The common thread and the insights gained are theoretically easy to follow. In practice, the captions do not seem to match the numbering or references in the text. In line 164, for example, reference is made to Figure 4 - the content of which, in turn, does not match the explanations. Could there be a systematic error here? Please check and clarify if necessary.
Author Response
Dear Reviewer
thank you for your comments. We are uploading the revised manuscript and our point-by-point response to the comments.
Please see the attachment.
Best regards
Chiara Parretti et al.

Reviewer 3 Report
Thank you for the possibility of reading this paper. It deals with the problem of the information systems and their appropriate use. I believe that the paper and the problem emphasized may be interesting for researchers who deal with this special area of knowledge.
Despite this, I have a few suggestions and comments.
- The motivation should be more highlighted, as well as the value-added of the paper
- The section "Conclusions" should be extended. It should include the practical applications and policy recommendations, as well as the implication for policy and policymakers. The importance of the axiomatic design should be highlighted.
- The data protection should be more extended. Information systems are full of sensitive and valuable information. Thus that kind of information needs special protection. This issue should be highlighted in a stronger manner. Now, only a little information is presented by the Authors (page 2).
- There are mistakes related to the ordering and numbering figures on pages 11-12. The Authors should put attention to this issue. Moreover, Figure 6 is probably table, not figure. In the body of the paper, there is a lack of references to (current) Figure 6 and Figure 7. Instead of this, the Authors have written about Figures 4 and 5. Please check and correct this part of the paper.
- The reference style and reference list should be corrected
- Some editorial mistakes in the middle paragraph on page 10 (lack of dot, additional spacebar – see lines 348-349 for example
- Figure 3 probably should be replaced with a table
- Different styles in bullet lists on pages 5-6, as well as 8-9.
Author Response

(The authors gave the same response as above.)
